# Fabrication of Magnesium-Aluminum Composites under High-Pressure Torsion: Atomistic Simulation

**Polina Viktorovna Polyakova [1,\*], Julia Alexandrovna Pukhacheva [2], Stepan Aleksandrovich Shcherbinin [3], Julia Aidarovna Baimova [1,2]** and **Radik Rafikovich Mulyukov [1,2]**

1   Institute for Metals Superplasticity Problems of RAS, St. Khalturina, 39, 450001 Ufa, Russia; julia.a.baimova@gmail.com (J.A.B.); radik@imsp.ru (R.R.M.)
2   Department of Physics and Technology of Nanomaterials, Bashkir State University, St. Zaki Validi, 32, 450076 Ufa, Russia; coldday87@gmail.com
3   Peter the Great St. Petersburg Polytechnic University, Polytechnicheskaya, 29, 195251 St. Petersburg, Russia; stefanshcherbinin@gmail.com
*   Correspondence: polina.polyakowa@yandex.ru

**Abstract:** The aluminum–magnesium (Al–Mg) composite materials possess a large potential value in practical application due to their excellent properties. Molecular dynamics with the embedded atom method potentials is applied to study Al–Mg interface bonding during deformation-temperature treatment. The study of fabrication techniques to obtain composites with improved mechanical properties, and dynamics and kinetics of atom mixture are of high importance. The loading scheme used in the present work is the simplification of the scenario, experimentally observed previously to obtain Al–Cu and Al–Nb composites. It is shown that shear strain has a crucial role in the mixture process. The results indicated that the symmetrical atomic movement occurred in the Mg–Al interface during deformation. Tensile tests showed that fracture occurred in the Mg part of the final composite sample, which means that the interlayer region where the mixing of Mg, and Al atoms observed is much stronger than the pure Mg part.

**Keywords:** composite; molecular dynamics; magnesium; aluminum; mechanical properties





## 1. Introduction

Magnesium (Mg) alloys are of considerable interest nowadays in the automotive and aerospace industry since they have lightweight properties[1,2]. Moreover, magnesium-based alloys can be biocompatible and biodegradable. However, magnesium has quite low strength, elastic modulus, creep resistance, and formability [3], which considerably limits its applications. Thus, the search for new possibilities to fabricate composites based on Mg and other metals that can demonstrate improved mechanical properties is of high importance. For example, aluminum (Al) can prevent the corrosion process of Mg alloy and possess better properties for both Mg and Al [4–6].

One of the possible ways to obtain metallic composites is high-pressure torsion (HPT), during which high compressive and shear stresses result in the formation of the composite structures. To date, several types of composite materials were obtained by HPT from metallic plates: Al–Cu [7–13], Al–Mg [14–16], Al–Nb [17], and Al–Ti [18]. Since severe plastic deformation can increase the diffusion in the materials [19,20], in situ composites can be obtained by HPT through the bonding of different metals, which can lead to the formation of new intermetallic phases. According to the phase diagram, several intermetallic Al–Mg phases can be obtained, which are AlMg, $Al_3Mg_2$-$\beta$, $Al_{30}Mg_{23}$-$\varepsilon$, and $Al_{12}Mg_{17}$-$\gamma$ [21]. The intermetallic $Al_{12}Mg_{17}$ can significantly influence the corrosion and mechanical properties of the Al–Mg structure.

Fundamental aspects of deformation behavior between Al and Mg and the formation of intermetallic compounds can be effectively studied by molecular dynamics (MD). Due

to some limitations of experimental methods, MD is widely used to study phase transformations and to determine structural properties of different materials [22–25]. Moreover, it allows scientists to visualize the structure on the atomistic level, analyze the distribution of atoms during different processing stages, and calculate physical or mechanical properties. MD was used for studying of effect of grain boundary segregation on the deformation mechanisms and mechanical properties of Al–Mg alloys [26–28], reduction of tensile strength for Al matrix with Mg inclusion [29], strengthening effects of basal stacking faults in Mg [30], etc.

In this work, the interactions between Al and Mg on Al–Mg interface under compression combined with shear load using the MD simulation method is studied. This combination of compression and shear is the attempt to realize HPT which was successfully used to obtain in situ composites. The present work is the continuation of the previous work by authors of [31], in which preliminary results were published. To study the strength of obtained mixed structure on the Al–Mg interface, a tensile strain is applied to the sample.

## 2. Computational Methods

To study the process of atomic mixing under deformation treatment near the interface of different metals, MD modeling of Mg–Al sample, containing an interfacial boundary was used. The cubic sample contained 54,170

atoms and was a simulation box with a dimension of $10.0 \times 10.0 \times 10.0$ nm$^3$. The contact surfaces of Mg and Al were (0001) and (001) planes, respectively. The Al and Mg layers were 5.0 nm thick. The schematic of the initial structure is shown in Figure 1, where the face-centered cubic (FCC) Al is bordered by the hexagonal (HPC) Mg. The atomic radius of Al was 143 pm and for Mg was 160 pm; atomic masses were 26.98 for Al and 24.307 for Mg. The interlayer distance between two crystals was calculated as $(a_{Mg} + a_{Al})/2$ = 3.6 Å. Periodic boundary conditions were applied along $x$, $y$, and $z$. Melting temperatures of Al and Mg were close and equal to 660 °C and 650 °C, respectively.

The sample with the minimized interfacial boundary energy was used for modeling. To obtain the initial structure in an equilibrium state, the minimal energy of the system was achieved by the multiple corrections of the atomic positions with the help of the steepest descent method, terminated if the variation in the energy or force was less than a given value. The initial structure was relaxed until the local or global minimum of the potential energy was reached. The simulation run was terminated when one of the stopping criteria (energy or force) was satisfied. The main goal of the relaxation process, in this case, was to obtain equilibrium stresses on the interface. After several numerical experiments with different parameters of minimization, stopping tolerance energy $10^{-24}$ and stopping tolerance force $10^{-26}$ eV/Å were chosen. In the present case, the setting of tolerance force $10^{-26}$ means no $x$, $y$, and $z$ component of the force on any atom will be larger than $10^{-26}$ eV/Å after minimization.

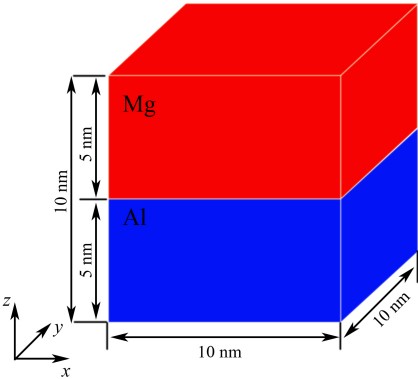

**Figure 1.** Schematic of the initial sample.

After the minimization process at 300 K, the simulation box was initially stress-free. The equilibrium state implies that normal stresses $\sigma_{zz}$ in the two crystals are zero; the summation of stress $\sigma_{xx}$ in the two crystals and the summation of stress $\sigma_{yy}$ in the two crystals are zero. Moreover, the two dimensions of the sample ($x$ and $y$) were not arbitrarily chosen because of the incommensurate nature of the FCC and HCP crystals but were determined such that the strains imposed on Al and Mg semi-infinite perfect crystals were minimized, ensuring periodic boundary conditions and equilibrium of the initial structure.

The simulations were carried out by MD using the large-scale atomic/molecular massively parallel simulator (LAMMPS) package. For temperature control, the Nose–Hoover thermostat was applied. Verlet algorithm to integrate the Newtonian equation of motion with an integration time-step of 2 fs was used. The well-approved Mg–Al embedded atom method (EAM) potential [32] was used in the molecular dynamics model. The EAM potentials for Al and Mg have been constructed on the basis of the experimental data and successfully used for a previous simulation in [33].

To obtain the composite structure and mixing of the atoms near the Al–Mg interface, uniaxial compression normal to the Al–Mg interface ($\varepsilon_{zz} < 0$), combined with shear in the interface plane ($\varepsilon_{xy}$ in this case), was applied. It should be noted that in accordance with the deformation mechanism inserted in the LAMMPS simulation package, shear deformation is simply the change of the $xy$, $xz$, and $yz$ tilt factors of the simulation box with the given strain rate [34]. To apply compression deformation, change of the specified dimension of the box via constant displacement (in this case, it was applied along $z$-dimension), which is effectively a constant engineering strain rate, occurred.

As previously shown [31,33], pure compression is not efficient enough to obtain composite. Thus, in this study, compression was combined with shear to reproduce analog to high-pressure torsion. Strain rates were $\dot{\varepsilon}_{zz} = 6.2 \times 10^{-8}$ ps$^{-1}$ and $\dot{\varepsilon}_{xy} = 6.2 \times 10^{-7}$ ps$^{-1}$. Numerical experiments were conducted at room temperature 300 K, since it is the usual temperature for experiments.

In Figure 2, the stress–strain curve of the sample under compression is presented, along with the snapshots of the structure at three strain stages. There are several different possibilities to present stress–strain curves and calculate equivalent stress and strain in experiments when HPT is conducted [35–38]. However, the most useful equations for calculations of equivalent stress and strain proposed previously are not always suitable [38], especially when considering such a simple simulation, in which just two strain components are realized. For the present model, recalculation of compressive and shear stress and strain to the equivalent one can become physically unreasonable. Thus, stress–strain curves were presented for both components. Further, the shear strain is mentioned as characteristic for the description of the obtained results, since shear strain, in particular, results in better mixing of atoms.

The course of the curve for compression (see Figure 2a) before step II is almost linear, without considerable stress fluctuations, in comparison with the curve after step III. As it can be seen from the snapshots of the structure, even at $\varepsilon_{xy} = 0.5$ ($\varepsilon_{zz} = 0.05$), very good mixing of the atoms occurred, However, the strength of the interface and microstructure peculiarities should be estimated. In comparison with compressive components, shear stress is low (about 1 GPa), while the achieved strain is 10 times higher.

To check the strength of the obtained composite, tensile loading normal to the Al–Mg interface was applied ($\varepsilon_{zz} > 0$ in this case). Since this work is an attempt to reproduce the experimental work, all the parameters were chosen to be close to the experimental. Thus, tensile numerical tests were conducted at 300 K, which is usual for experiments. The tensile strain imposed in this simulation was performed by deforming the simulation box. During the dynamic loading, the stress was attained by the averaged stress, and the strain was derived from the positions of the periodic boundaries along the $z$-axis.

Tensile strain was applied after three steps of deformation and to the initial structure. The structure of the composite, obtained after compression to stage I, was considered as initial for the tensile test and was not additionally relaxed or changed. The same principle

was applied for structures after compression to stage II and III. Thus, structures at points I, II, or III in Figure 2 are considerably stressed and at non-equilibrium.

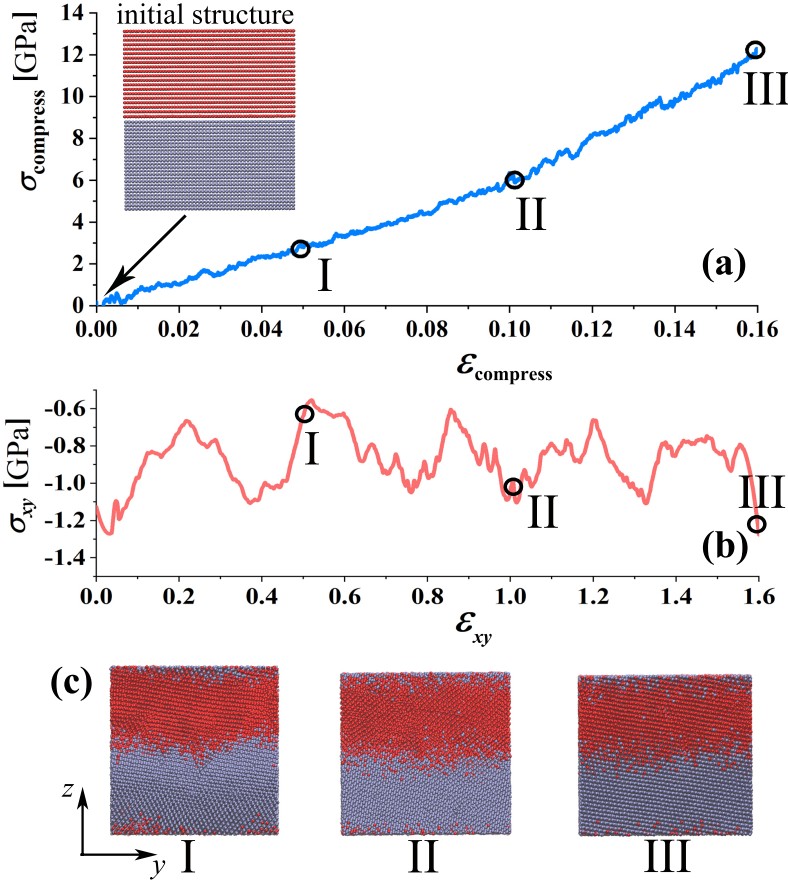

**Figure 2.** Stress–strain curves during uniaxial compression normal to the Al–Mg interface combined with shear: (**a**) compressive strain and stress; (**b**) shear strain and stress; (**c**) the snapshots of the structure at three stages of deformation. The red color is for Mg atoms, the blue color is for Al atoms.

Visualization of MD simulation data and structure analysis were carried out using the VMD [39] and OVITO [40] tools.

## 3. Results

### 3.1. Composite Fabrication

During deformation treatment, considerable mixing of Al and Mg atoms occurred through the interface. In Figure 3, changes of the atomic positions through Al–Mg interface for $0.0 < \varepsilon_{xy} < 1.6$ are presented. Mg block in the model is shifted to the right by about 100 Å for a clearance. As it can be seen, mostly, atomic mixing occurred during the first deformation stages $\varepsilon_{xy} < 0.4$. The first atomic movement occurred at $\varepsilon_{xy} = 0.017$ simultaneously with the appearance of base-centered cubic (BCC) lattice defects in the Mg part of the sample. Atomic positions between $\varepsilon_{xy} = 0.027$ and $\varepsilon_{xy} = 0.4$ are not presented since the continuous atomic movement occurred: Al atoms move towards the Mg part of the sample and vise versa.

The process of atomic migration can be better described by the average and maximum distances of an atomic displacement in comparison with the initial position of the boundary, which is presented in Figure 4. The value of $\Delta z$ is calculated as the average movement of the atoms from the initial position of the interface. At $\varepsilon_{xy} = 0.017$, Al atoms moved from the atomic planes closest to the interface for $\Delta z_{max} = 2.2$ Å, and Mg atoms at the same strain moved for $\Delta z_{max} = 2.3$ Å. Considerable displacement of Al (Mg) atoms inside Mg (Al) part

of the sample occurred before $\varepsilon_{xy}=0.4$, with the average displacement for 4 Å, while for $0.04 < \varepsilon_{xy} = 1.6$, the average atomic displacement is about 2 Å.

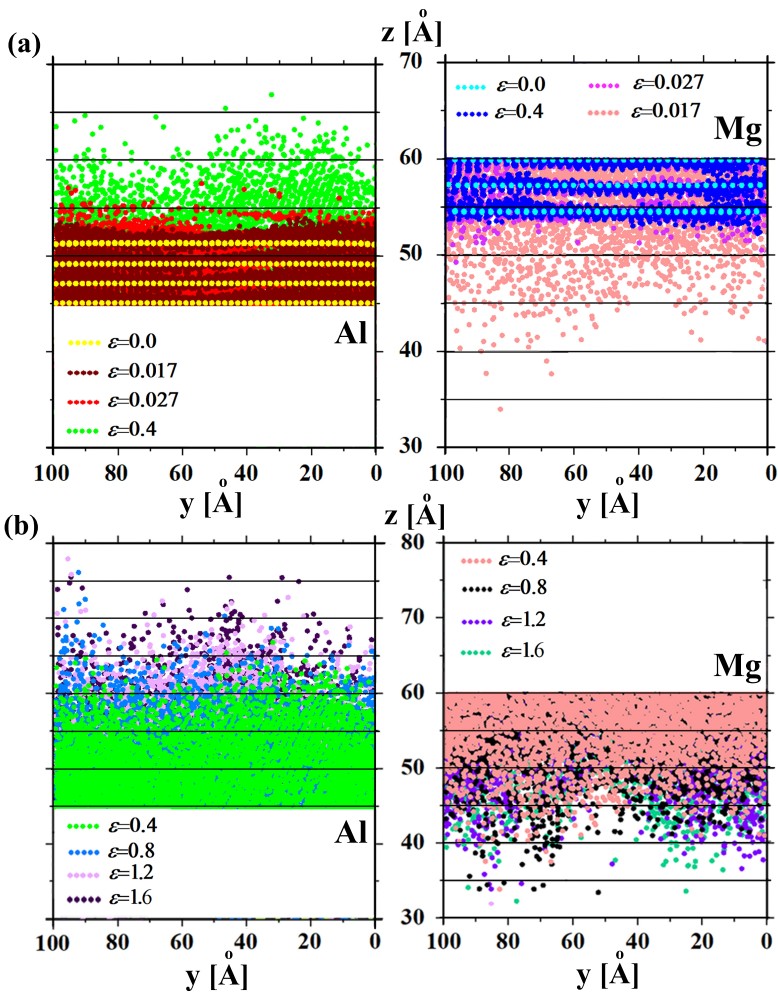

**Figure 3.** Changes of the atomic positions through Al–Mg interface: (**a**) for $0.0 < \varepsilon_{xy} < 0.4$ and (**b**) for $0.4 < \varepsilon_{xy} < 1.6$. Different colors correspond to different deformation stages. Only part of the sample is presented.

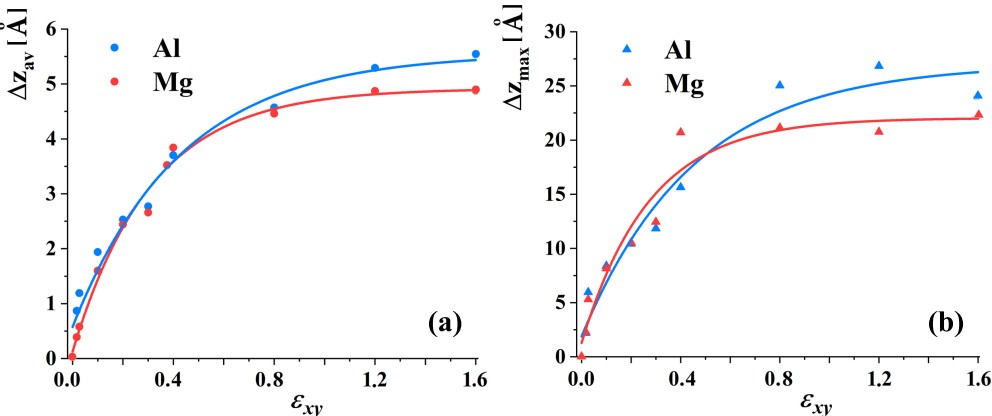

**Figure 4.** Changes of the atomic positions $\Delta z$ as the function of compression strain. (**a**) $\Delta z_{av}$, (**b**) $\Delta z_{max}$.

As it can be seen from Figure 3, both average distances and changes of the maximum atomic displacements can be described by the power function for both metals. It is well seen that $\varepsilon = 0.4$ is enough to obtain good mixing of the atoms near the interface. After $\varepsilon = 1.6$, compression strain begins to prevent mixing, since the structure near the boundary is strongly compressed. As it is found, Al atoms move towards the Mg part of the sample slightly better, which is connected with the difference of the lattices: Mg HPC lattice is not as densely packed as FCC Al lattice. However, it should be mentioned that the number of Mg atoms moving into the Al part is almost equal to the number of Al atoms moving into the Mg matrix. This can be explained by the similarity of the atomic radii of Mg and Al. The lack of difference in the melting point for Mg (923 K) and Al (933.5 K) is worth considering, which means the bonds in both metals are similarly stronger to fracture. For both metals, a symmetrical binding movement occurred during the deformation of the initial sample.

In accordance with common-neighbor analysis (CNA), during the first stage of deformation (until $\varepsilon_{xy} = 0.4$), the Al part of the sample preserves FCC lattice, and the Mg part of the sample preserves HCP lattice with the appearance of BCC lattice. After $\varepsilon_{xy} = 0.0015$, a considerable number of dislocations appeared in the Al part, about two times more than in Mg. Dislocations in the Mg part rapidly move and disappear during shear, while in Al, they change type but never entirely disappear. Numerous dislocations appeared on the interface during a mixture of Al and Mg atoms.

For the first chosen stages I and II (before $\varepsilon = 1.0$), in the Mg part, mostly the HCP lattice can be seen from CNA, but further, the share of the BCC lattice increases, which is the main difference between structure at stage I and II, in comparison with stage III.

Although mixing dynamics is quite similar, structural transformations during compression to $\varepsilon_{xy} = 0.5$, $\varepsilon_{xy} = 1.0$, and $\varepsilon_{xy} = 1.6$ are different. The structure obtained after stages II and III are similar, in comparison with structure after stage I, which means that the strength of the final composite would be different.

In the experiment [9,10,12,13,17], annealing at 450 °C is applied after the initial deformation of the Al–Cu composite. It is observed, that after annealing, two more intermetallic phases have appeared in the structure, and microhardness increase two times. Although this work is also based on the experiment on Al–Mg composite fabrication, there are no published results for Al–Mg annealed after HPT.

In the present work, annealing at temperatures between 250 and 450° was conducted. However, no noticeable effect is observed for all the considered temperatures. Limitations of the MD model do not allow us to simulate annealing since considerable time is required for this process. Even four times bigger annealing time is not enough.

*3.2. Al–Mg under Tension*

In Figure 5, stress–strain curves with characteristic marks during tensile loading normal to the Al–Mg interface are presented for the tension of the initial sample (black curve, 1) and tension after compression at stages I (red curve, 2), II (green curve, 3), and III (blue curve, 4). Several pop-in events are observed on the stress–strain curves, which can be attributed to the release of strain energy accumulated during the deformation through defect activities. To analyze deformation mechanisms and strength of the composite, snapshots of the structure during tensile loading are presented for curves 1–3 in Figure 6, in accordance with CNA, and for curve 4 in Figure 7, in accordance with CNA and dislocation analysis.

The first structural changes of the initial sample occurred at $\varepsilon = 0.03$ in the Mg part of the sample (see Figure 6a). Fracture is observed in the Mg part of the sample at $\varepsilon = 0.045$ (shown by a circle in Figure 6a). Close to this limit, dislocations appeared in Mg together with other structural defects. This result is quite expected since Mg is well known for having low strength. However, this result is important for further comparison.

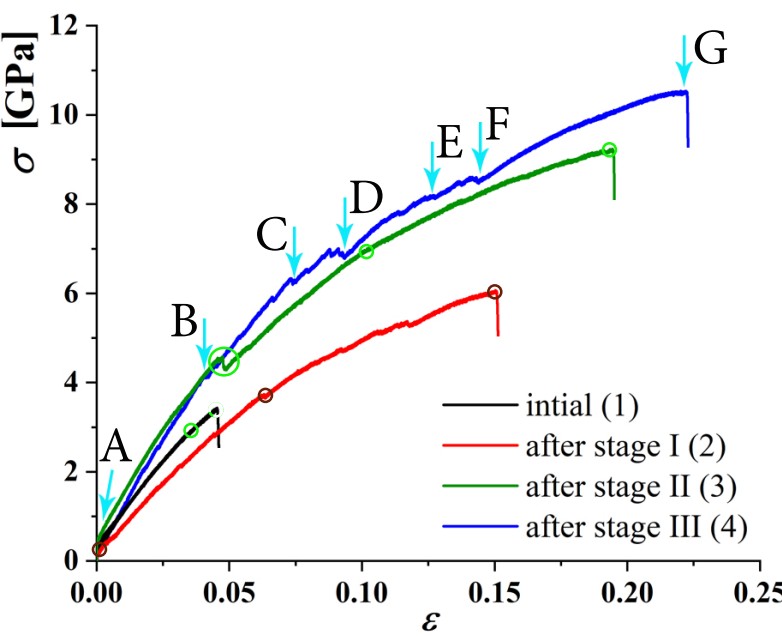

**Figure 5.** Stress–strain curves during tension normal to the interface region after different initial compression strain: after $\varepsilon_{xy} = 0.5$ (2); $\varepsilon_{xy} = 1.0$ (3), and $\varepsilon_{xy} = 1.6$ (4).

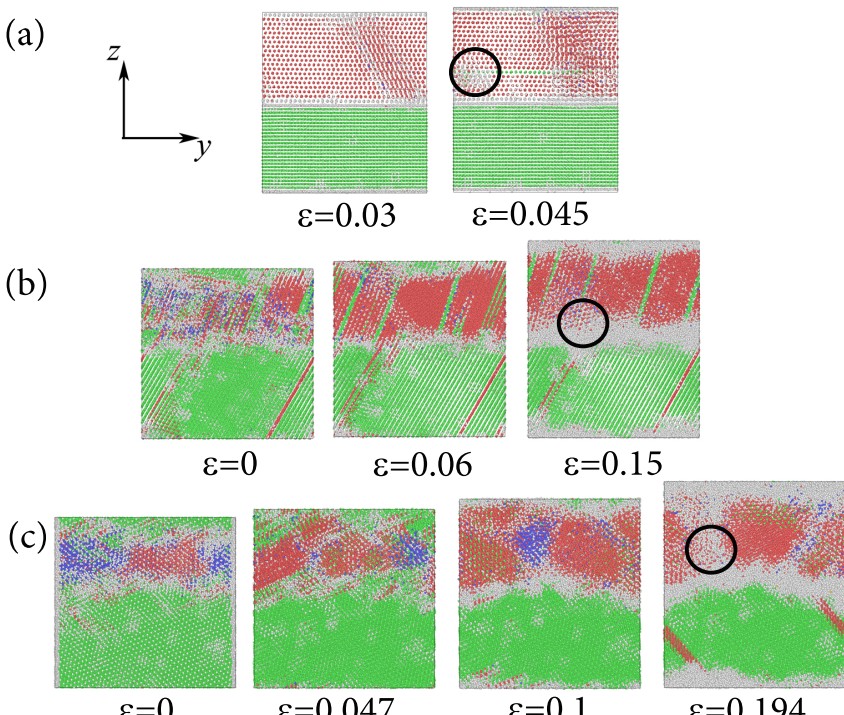

**Figure 6.** Snapshots of the structure in accordance with stress–strain curves shown in Figure 5: (**a**) tension of the initial sample; (**b**) tension of the sample after compression, stage I; (**c**) tension of the sample after compression, stage II. All atoms are colored by the CNA parameter, where HCP atoms are red, FCC atoms are green, BCC atoms are blue, and other atoms are gray.

From Figure 6b, it can be seen that fracture of the sample compressed to stage I occurred at $\varepsilon = 0.15$ on the boundary between Al and Mg, where the mixed structure is obtained. This means that the interlayer part is even weaker than the Mg part (see also Figure 5, curves 1 and 2). At tensile strain $\varepsilon = 0.0$, there are two lattices in the MG part of the sample—HPC (red atoms) and BCC (blue atoms); however, after $\varepsilon = 0.06$, BCC phase almost disappears. It should be noted that a single HCP atom layer represents a twin

boundary, two adjacent HCP atom layers present an intrinsic stacking fault, and an FCC atom layer in the middle of two HCP atom layers stands for an intrinsic stacking fault. As in the experiments or other MD simulations, twinning is one of the effective mechanisms of deformation for Mg [41,42].

From Figure 6c, it can be seen that fracture of the sample compressed to stage II occurred at $\varepsilon$ = 0.194 in the Mg part of the sample, between two boundary regions. At $\varepsilon$ = 0.047, the share of the BCC phase considerably decreases; however, the Mg part of the sample contains the BCC phase during the whole deformation process.

Figure 7 presents a series of snapshots exhibiting the defect evolution during tension after stage III of compression corresponding to the characteristic points marked in the stress–strain curve 4 of Figure 5. This case is chosen for detailed analysis because the strength of the sample after stage III is the highest. The other atoms are colored grey, the HCP atoms are colored red, BCC atoms are colored blue, and the FCC atoms are colored green. In accordance with the OVITO dislocation extraction algorithm, the stair-rod dislocation lines are colored purple, the Hirth dislocation lines are colored yellow, Frank dislocation lines are colored light blue, the Perfect dislocation lines are colored blue, and the Shockley dislocation lines are colored green.

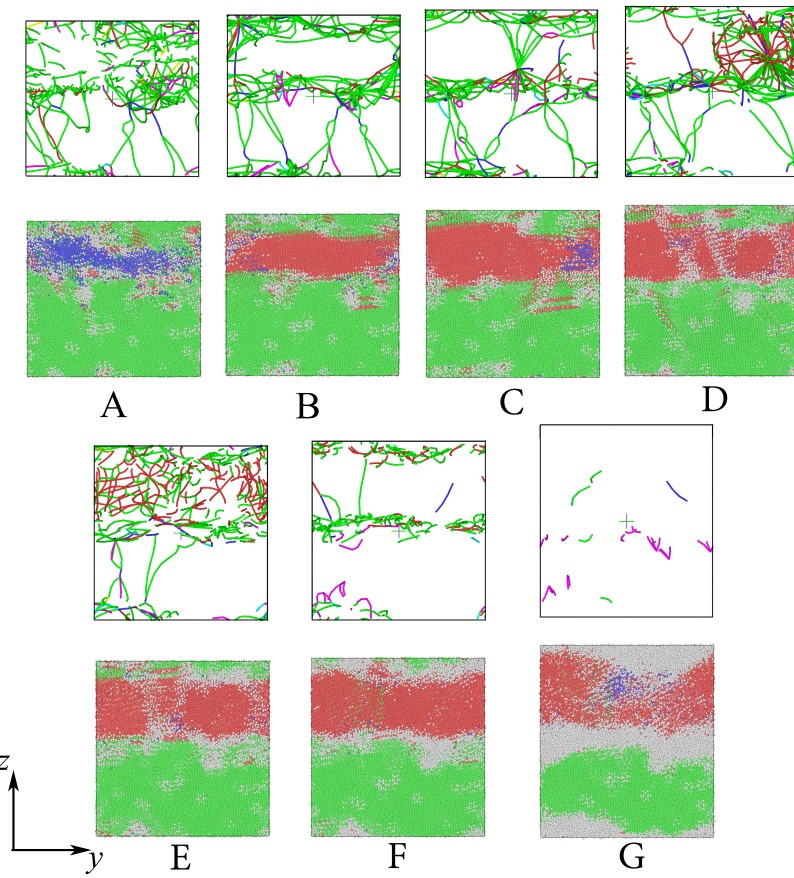

**Figure 7.** Snapshots of the structure in accordance with stress–strain curve shown in Figure 5 for tension after stage III. All atoms are colored by the CNA parameter, where HCP atoms are red, FCC atoms are green, BCC atoms are blue, and other atoms are gray.

After compression to $\varepsilon$ = 0.16, the BCC phase is dominant in the Mg part of the sample. It should be mentioned that an even boundary region with mixed Al and Mg atoms is shown by green FCC lattice. At tensile strain $\varepsilon$ = 0.04, the BCC phase almost disappears, and this region on the stress–strain curve is almost linear. Curves 3 and 4 from Figure 5 coincide before $\varepsilon$ = 0.045, and both have similar structural transformations. Numerous dislocations appeared on the interface between Al and Mg.

At point C, new dislocations appeared in the Mg part, while dislocation distribution in Al remained almost the same from $\varepsilon = 0.05$ to $\varepsilon = 0.1$. After $\varepsilon = 0.1$, dislocation network is developed, mainly in the Mg part, until a tensile strain equal to 0.15 is achieved. After that, as it can be seen from Figure 5, on curve 4, no pop-in events are observed, which is connected with the total change of the dislocation structure (Figure 7F,G).

Fracture occurred at $\varepsilon = 0.22$ on the opposite side of the crystal, which cannot be seen in Figure 7G, but the crack appeared in the Mg part of the composite, where the BCC phase is localized. Close to the strength limit, almost all dislocations disappeared from the structure.

## 4. Conclusions

Molecular dynamics simulation was used to study and analyze atomic movement on the Al–Mg interface under high pressure combined with shear strain. The proposed model was based on the scenario, experimentally observed previously in [9,10,12,13,17] for Al–Nb and Al–Cu composites. It is found that compression combined with shear is an effective method to obtain composite structure on the Al–Mg interface.

Considerable mixing of Al and Mg atoms occurred after compressive strain 0.04, with simultaneously applied shear strain 0.4; however, it is found that at this stage, the mixed Al–Mg region is weak, and fracture occurred on the boundary region. Further deformation treatment is required to obtain the formation of a strong interface. From the tension tests, it is found that there is a critical compression level after which no considerable structural changes can be achieved. Moreover, the higher the applied shear strain is, the higher the strength of the composite is.

In the present work, no effect is observed after annealing of the compressed samples, which is in contradiction with previous experimental results on the Al–Cu interface [9,10,12,13,17]. This can be explained by the difference in the metals used for composite fabrication, the weaknesses of the methodology, or by the lack of understanding of how to find better annealing temperatures. The temperature of 450 °C chosen from the experiments for Al–Cu does not facilitate the mixing of the atoms.

**Author Contributions:** Conceptualization, J.A.B. and R.R.M.; methodology, P.V.P.; formal analysis, P.V.P.; resources, S.A.S.; investigation, J.A.P.; writing—original draft preparation, J.A.B. and R.R.M. All authors have read and agreed to the published version of the manuscript.

**Funding:** This research was funded by Russian Science Foundation Grant Number 18-12-00440. Work of R.M. supported by the program of fundamental researches of Government Academy of Sciences of IMSP RAS. The work of S.S. was supported by the Government of the Russian Federation (state assignment 0784-2020-0027).

**Acknowledgments:** The authors wish to acknowledge Peter the Great Saint-Petersburg Polytechnic University Supercomputer Center "Polytechnic" for computational resources.

**Conflicts of Interest:** The authors declare no conflict of interest.

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
