# Peer review of "Fabrication of Magnesium-Aluminum Composites under High-Pressure Torsion: Atomistic Simulation"

_applsci, doi:10.3390/app11156801_

Round 1
Reviewer 1 Report
The manuscript reports molecular dynamics results on the aluminum-magnesium (Al-Mg) composite materials. This material has a large potential for practical applications. Molecular dynamics was based on the embedded atom method and LAMMPS, the results may be of interest for some readers. The paper is well organized, but English needs to be substantially improved.
My main concern comes from the effect of annealing which was calculated for a wrong temperature value. As a result, the authors obtained no effect after annealing of the compressed samples in contradiction with the previous experimental results on the corresponding Al/Cu interface. It implies that the simulation must be repeated for the lower annealing temperature for the Al/Mg interface or removed as irrelevant.
Therefore, I recommend revising the manuscript.
Author Response
Thank you very much for kindly editing our manuscript and giving us the opportunity to revise it.
We are very grateful for the great efforts and valuable suggestions made by the Reviewers. We have revised the manuscript accordingly. The detailed response to the comments of the Reviewers is provided below together with the description of the changes applied to the manuscript. We note that all changes to the manuscript are highlighted in red for the convenience of the reviewer.
We wish that our revised manuscript and responses to the reviewers’ comments could be satisfactory. Thank you very much for your time and consideration.
Sincerely yours,
Polina V. Polyakova
Reply to Reviewer 1
Comment 1:
The paper is well organized, but English needs to be substantially improved.
Reply:
We thank the referee for such a high estimation of the organization of our work. We have tried to improve our English writing.
Comment 2:
My main concern comes from the effect of annealing which was calculated for a wrong temperature value. As a result, the authors obtained no effect after annealing of the compressed samples in contradiction with the previous experimental results on the corresponding Al/Cu interface. It implies that the simulation must be repeated for the lower annealing temperature for the Al/Mg interface or removed as irrelevant.
Reply:
We are extremely grateful to Reviewer #1 for pointing out this problem. Additional calculations were done for the study of annealing. We have checked not only several other temperatures but also the effect of annealing time. However, there is no effect of annealing in the present model. It can be connected with the weakness of the potential, but more likely it is connected with the fact that structure is considerably compressed at this stage and diffusion is blocked. While during shear strain mechanical mixing of the atoms took place. As well this work is based on the experiments for Al/Mg composite (which are not published yet) we discuss the effect of annealing (for two temperatures in the revised manuscript). But in accordance with the referee's advice, we removed Fig. 5 and also section “Annealing”. We rewrite the description of the annealing process.
Reviewer 2 Report
In their manuscript, the authors describe a series of MD simulations to study the high-pressure torsion process on a Al-Mg bicrystal. The method section of this article is insufficient because many important details of the setup of the MD simulations have not been provided: What are the crystallographic orientations of the two crystals? How was the equilibration process conducted? Was the box initially stress-free? How large are the mismatch strains at the interface? What is the temperature during HPT and tensile testing? How is shear strain applied in the periodic box? What are the boundary conditions for the tensile testing (uniaxial stress or uniaxial strain)? Without providing such details, the quality of the results cannot be judged properly. However, there seem to be some issues with the consistence, because in Figure 6 it is seen that the initial stiffness of predeformed samples is higher than that of the prestine bi-crystal. This typically indicates to some problems in the equilibraion process or to high internal stresses in the box.
Another major point for criticism is the statement that the authors interpret their results in terms of diffusion. How did the distinguish between diffusion and mechanical mixing? Typically this could only be done be running the simulations at various temperatures and judging the thermal activation of the process. In MD simulations it is notoriously difficult to draw valid conclusions on diffusion because of the time scale that is very far away from that of diffusive processes.
Some smaller points are that the results should be plotted as shear stress over shear strain or, even better, as equivalent stress over equivalent strain. What is the motivation to raise the compressive stress continuously? Isn't the real HPT process conducted under a constant compressive stress?
In the axis labels, units should be provided in brackets, not separated by a comma.
Author Response
Thank you very much for kindly editing our manuscript and giving us the opportunity to revise it.
We are very grateful for the great efforts and valuable suggestions made by the Reviewers. We have revised the manuscript accordingly. The detailed response to the comments of the Reviewers is provided below together with the description of the changes applied to the manuscript. We note that all changes to the manuscript are highlighted in red for the convenience of the reviewer.
We wish that our revised manuscript and responses to the reviewers’ comments could be satisfactory. Thank you very much for your time and consideration.
Sincerely yours,
Polina V. Polyakova
Reply to Reviewer 2
Comment 1:
The method section of this article is insufficient because many important details of the setup of the MD simulations have not been provided. Without providing such details, the quality of the results cannot be judged properly.
Reply:
We appreciate the comment. Definitely, the lack of these important details is a great disadvantage. We are sorry for the unclear presentation. Additional descriptions were added to the section “Simulation details”. A detailed reply is also presented below.
Comment 1.1 What are the crystallographic orientations of the two crystals?
Reply: The contact surfaces of Mg and Al are (0001) and (001) plane, respectively.
Comment 1.2 How was the equilibration process conducted?
Reply:
To obtain the equilibrium state of the considered structures, the energy of the system was minimized by the multiple corrections of the atomic positions by the steepest descent method, terminated if the variation in the energy or force is less than a given value. All the structures are relaxed until the system reaches its local or global minimum of potential energy. The simulation run is terminated when one of the stopping criteria (energy or force) is satisfied. The main goal of the relaxation process, in this case, is to obtain equilibrium stresses on the interface. After several numerical experiments with different parameters of minimization, stopping tolerance energy 10^{-24} and stopping tolerance force 10^{-26}eV/Angstrom is chosen. Stopping tolerance energy is dimensionless and met when the energy change between successive iterations divided by the energy magnitude is less than or equal to the tolerance [53]. Specified force tolerance is given in force units since it is the length of the global force vector for all atoms. For example, in the present case, the setting of tolerance force 10^{-26} means no x, y, and z component of the force on any atom will be larger than 10^{-26}eV/Angstrom after minimization.
Comment 1.3 Was the box initially stress-free?
Reply: Yes, the simulation box was initially stress-free, since we used minimization procedure.
Comment 1.4 How large are the mismatch strains at the interface?
Reply: At first, size of the structure was chosen so, to decrease the stress and strain mismatch in the interface between two different layers. Also, it is chosen so, to avoid strains on the boundaries of the simulation box at the initial state. This is additionally described in the text.
Comment 1.5 What is the temperature during HPT and tensile testing?
Reply: Since in the experiment, on which this work is based, the temperature during HPT is equal to 300 K, in the present model, the same temperature is chosen. It should be noted, that in the experiment, a local temperature increase took place, which is not taken into account in the present model. But in the present work area of the interface is quite small for better understanding of atomic mixing and it can be assumed that over this area temperature is 300 K. Tensile tests also considered at 300 K, as well it is room temperature usual for such experiments. All the conditions were chosen in close connection to experiments (experimental results are still not published).
Comment 1.6 How is shear strain applied in the periodic box?
Reply: In accordance with the mechanism inserted in LAMMPS simulation package, shear deformation is simply the change of the xy, xz, yz tilt factors of the simulation box with the given strain rate.
Comment 1.7 What are the boundary conditions for the tensile testing (uniaxial stress or uniaxial strain)?
Reply: Uniaxial strain is applied. In accordance with the mechanism inserted in LAMMPS simulation package, change of the specified dimension of the box via “constant displacement” which is effectively a “constant engineering strain rate” took place. Thus, the strain imposed in tensile simulation is performed by deforming the simulation box, namely remapping the atom positions.
Comment 2:
However, there seem to be some issues with the consistence, because in Figure 6 it is seen that the initial stiffness of predeformed samples is higher than that of the prestine bi-crystal. This typically indicates to some problems in the equilibraion process or to high internal stresses in the box.
Reply:
We appreciate the comment. We are sorry for the unclear description. In fact, samples taken for tensile tests were not relaxed. The Initial Al/Mg sample was compressed until quite high strains. Then, data, collected during compression were divided into three regions in accordance with the course of the pressure-strain curve, but we just take the structural state, make it as the initial structure for tension, and made the calculations. No relaxation was applied before. Thus, structures at points I, II, or III in Fig. 2 are considerably stressed. From one point of view, we can apply relaxation, remove stresses and make the calculations. But in the present work, the aim was to study these stressed states of the structure. The initial stiffness of the structures after state I and initial undeformed structure could become different after relaxation. But in this case, the difference is because of the insufficient mixture of atoms near the interface. And we can conclude further that additional shear can result in much better atomic mixing. Additional description is added to the text.
Comment 3:
Another major point for criticism is the statement that the authors interpret their results in terms of diffusion. How did the distinguish between diffusion and mechanical mixing? Typically this could only be done be running the simulations at various temperatures and judging the thermal activation of the process. In MD simulations it is notoriously difficult to draw valid conclusions on diffusion because of the time scale that is very far away from that of diffusive processes.
Reply:
We are grateful for the reviewer’s criticism on this inappropriately used therm. We wrote in the text that MD not really good for the study of the diffusion process, but did not use this better explanation of atomic mixing. Of course, this is our mistake to call the observed process “diffusion”. In the revised version of the manuscript, we change the terminology and rewrite the text accordingly.
Comment 4:
Some smaller points are that the results should be plotted as shear stress over shear strain or, even better, as equivalent stress over equivalent strain. What is the motivation to raise the compressive stress continuously? Isn't the real HPT process conducted under a constant compressive stress?
Reply: We appreciate the comment. The reviewer is totally right in his suggestion. We used compressive stress for simplicity, but now we see that this is not the best way to present our results. After some literature review on this subject (new refs. [35-38]) we have found that there are several different possibilities to present stress-strain curves in experiments. However, the most useful equations for calculations of equivalent stress and strain proposed previously are not always suitable, especially when we are talking about such a simple simulation. If we did a continuous simulation of high-pressure torsion, then we should definitely use equivalent strain and stress. Also, we cannot say that we simulate exactly high-pressure torsion since we just simulate similar stresses. Finally, in the revised version, we decided to add stress-strain curves for shear components in addition to compressive components and explain this. We think that all the proposed equations for the calculation of equivalent strain are not really suitable for our model since are written for experimental works or theoretical descriptions. Thus, recalculated stresses and strain can become physically unreasonable. However, we provide an explanation of the text in accordance with the referee's advice.
Comment 5:
In the axis labels, units should be provided in brackets, not separated by a comma.
Reply: We appreciate the comment. Units are provided in brackets.
Round 2
Reviewer 1 Report
The authors addressed all my comments.
Minor issues with the manuscript:
1. Figure 4 is blank in the manuscript.
2. All full affiliations and emails should be given for the authors.
The manuscript can be published in Applied Sciences after these minor changes.
Author Response
We are very grateful for the great efforts and valuable suggestions made by the Reviewer. Fig. 4 is added to the manuscript. Full affiliations and emails are given for the authors.
Reviewer 2 Report
The authors responded to all issues in a proper way and ammended the manuscript accordingly.
Author Response
We are very grateful for the great efforts and valuable suggestions made by the Reviewer.